# Evaluation of the Professional Competence of Kitchen Staff to Avoid Food Waste Using the Modified Delphi Method

**Wen-Hwa Ko \*** and **Min-Yen Lu**

Department of Restaurant, Hotel and Institutional Management, Fu Jen University, 510 Chung Cheng Rd., Hsinchuang Dist., New Taipei City 24205, Taiwan; 407246025@gapp.fju.edu.tw
\*   Correspondence: 073770@mail.fju.edu.tw

**Abstract:** Avoiding food waste has become an important development trend nowadays, and the concept of sustainable cooking is becoming more and more important. How to integrate the concept of surplus food into the catering industry is, in reality, a topic of concern in all walks of life. This study discusses the indicators of professional competence that kitchen staff should have in cooking, from the perspective of surplus food. In this study, literature review, interviews, and the modified Delphi method were used to evaluate the importance of the indicators. After three rounds of the Delphi method analysis by 22 experts, the indicators of the professional culinary competence of kitchen staff, with regard to surplus food, were summarized in the results. The research results showed that there are a total of 58 indicators in professional culinary competence, which are divided into the following four dimensions: 13 items were knowledge competence indicators, 13 items were attitude competence indicators, 23 items were skills competence indicators and 9 items were problem and difficulty indicators. Among them, attitude was the most important indicator, followed by knowledge and skills.

**Keywords:** modified Delphi method; kitchen staff; food waste; professional competence

## 1. Introduction

Food waste is a global economic, ecological, and social problem. Every year, about 1.3 billion tons of food are lost or wasted globally, and the related costs amount to US$ 750 billion [1], which accounts for a large part of the total food production [2]. The key to the problem of food waste lies in the catering supply chain, because food costs and resources are invested in the production of meals, and they are then transported or processed to become catering commodities that are for sale. On the other hand, the catering industry and consumers should pay attention to this issue first, from both the supply and demand perspectives. Gobel et al. [3] suggested that crops that are not purchased by the distributors will be discarded, starting with the farmers, because their produce fail to meet the procurement standards in terms of the appearance, weight, shape, size, etc. Food will be directly discarded in the process of food processing, storage and retail, due to the improper use of machinery, time-consuming food processing, and poor bacterial control.

Reducing food waste is an important part of sustainable development, as it can provide an ample food system for future population growth and maintain the sustainable development of natural ecology [4,5]. Food waste occurs at all stages of the food supply chain [6], but the closer it is to the end of the supply chain, the greater the negative impact of food waste is on the environment, the economy, and society [7,8]. Curry [9] mentioned that food waste is a very important component of hospitality waste, accounting for more than 50% of it. A large amount of food is discarded in the process of food

preparation in restaurants, because it cannot be stored or re-used [10]. In a food waste research survey conducted in the United Kingdom, which examined 19 restaurant catering units [11], it was pointed out that the main cause of food waste was 45% due to the restaurant's improper meal preparation and 34% due to consumers leaving surplus food. Researchers on food waste in restaurants and group kitchen restaurants in Finland [12] and Sweden [10] found that food waste was caused by production depletion and surplus food, due to overproduction.

Since food waste has become a global issue, the chef is a front-line worker in the kitchen and plays a very important role in the process, from the purchasing to the final satisfaction of the consumers. Therefore, the problem of avoiding food waste should begin with the kitchen staff. Based on the related literature of professional culinary competence and the avoidance of food waste, this research developed an interview outline. After the experts responded to the interviews, the relevant evidence was extracted for consideration in the future education and planning of industry and academia, in order to improve the competence of food processing professionals and kitchen staff so that they can avoid food waste.

## 2. Literature Review

### 2.1. Food Waste

The Food and Agriculture Organization (FAO) [13] defined food waste as all food distributed to people that is discarded, damaged, despised, or that suffers from pests and diseases, throughout the food chain. The United States Environmental Protection Agency (USEPA) [14] believes that food waste refers to uneaten food from homes, businesses and institutions, or food that is discarded during food preparation, including what is done at home and in grocery stores, restaurants, bars, factory lunch rooms, company restaurants, etc.

WRAP (Waste and Resources Action Programme) [15] pointed out that there are two main reasons for food waste in British households, namely, excess food and food that is not consumed in time. Excess food refers to over-cooked dishes, the preparation of excessive ingredients, too much food on the plate and foods that are cooked and processed in excessive amounts, including the ingredients that are damaged during the cooking (e.g., scorched meat steaks), which are collectively referred to as "kitchen waste". Food that is not eaten up in time refers to food has not been eaten within the labeling period, such as the "expiration" and "best before" dates, resulting in moldiness, spoilage, or expiration. Parfitt et al. [2] noted that discarded food is a huge waste and that food discarded after purchase and cooking is of no use to anyone. These foods have been produced, packaged, transported, and stored, which have consumed energy, raw materials, water, and packaging. Food that is thrown away, when it is not eaten, not only wastes money and resources, but it also shows that the food that has been produced and processed has greatly exceeded the amount of food sold and eaten, which has a considerable negative impact on the environment. Richter [16] found that there are several factors that affect food waste, including the following: differences in the perceptions of waste, the influence of an ethical attitude, the perception of the food provider's role, inconvenience in purchasing, poor purchasing habits, the lack of planning, improper food cooking, the attitude towards surplus food, insufficient awareness, and no literacy on food waste. Visschers et al. [17] pointed out that the focus of food waste is on avoidable food waste and food or beverages that can be completely consumed or discarded because they are no longer needed, such as left-overs after a meal. The retail and consumption stages have the most potential for improving the reduction of food waste [2].

### 2.2. Catering Professional Competence

Professional competence is defined as the degree to which employees can apply it to their work conditions, their professional knowledge, skills, motives, and traits [18]. It includes high-level skills, such as critical thinking, teamwork, communication, and learning [19]. Ko [20] pointed out that professional catering competence that is composed of positivity and endurance is the most

important. However, management competence extends to a person's responsibility, cost calculations, a willingness to accept new knowledge, a positive attitude, and communication skills. The basis of professional functioning requires knowledge, the correct attitude, and skills. Ko and Hsiao [21] showed that culinary knowledge is the most important factor of the major alternatives, followed by personal characteristics and communication. Shum et al. [22] divided catering competence into three parts, namely: (a) leadership competence, which includes planning and organizing, analyzing and solving problems, demonstration and coaching, and information innovation; (b) personal competence, which includes expression, emotional intelligence, ethical concepts, the diversification of abilities, active learning, and effective communication; and (c) management competence, which includes team management, as well as teaching and guiding others. Harrington et al. [23] suggested that the main professional competencies of chefs include cooking skills, product knowledge, communication skills, sales marketing, and food concepts. Hu [24] mentioned that innovative culinary competence includes self-imagination, knowledge, affection, as well as sensory and aesthetics literacy. Suhairom et al. [25] stated that professional knowledge, personal qualities, and technical abilities are required to achieve culinary creativity.

*2.3. Current Status of Food Waste in the Catering Industry*

Considering the final consumption of production, retailers have the highest proportion of the food supply chain [26]. Beretta et al. [27] pointed out that the end of the food supply chain accounts for 40% of the food loss. Pirani and Arafat [28] discussed reducing food waste in the catering industry, and found that the most wastage occurs in buffets. In the catering sector in the UK, 45% of the food wasted comes from improperly-prepared meals in restaurants, and 34% comes from surplus food left by consumers [11]. Stockli et al. [29] mentioned that food waste in restaurants includes excessive ordering, using plates that are too large, etc., and that it could be avoided by advising consumers to take away their left-overs and to give discounts for having no leftovers. Carlos et al. [30] pointed out that fast-food restaurants or takeaways can adjust their menus and prices according to consumers' preferences. As can be seen from the above, if the catering industry asks its consumers to be aware of food waste, it will be reduced.

## 3. Methodology

*3.1. Design of the Modified Delphi Method (MDM) Questionnaire*

This study was designated to use the modified Delphi method (MDM) put forward by Murray and Hammons [31]. It replaced the test of the open-ended questionnaire with substantial relevant literature and deep interviews with experts, and developed a structural questionnaire to replace the first-round survey of the Delphi method.

This study employed a three-round MDM questionnaire. Round 1 consisted of deep interviews and a literature review for the development of a structured questionnaire. The importance of each standard was then evaluated by experts (with 5 representing the most important and 1 representing the least important), and open questions were designed for the experts to make suggestions. After collecting the questionnaires back and integrating and analyzing each item, it was modified to form a questionnaire for the next round. The competence of managers could be classified into three categories, according to the competence model proposed by Schoonover [32], namely: core competence, management competence, and professional competence. The competence of the kitchen staff was classified into core and professional competence.

Two or three rounds were the best for the Delphi method questionnaire, because if there were more than three rounds, the experts would feel tired and their willingness to cooperate would be reduced; however, the opinions given in the two or three rounds would be stable and consistent [33]. In this study, all the experts reached consensus after three rounds. The seven questions on the experts'

personal data included their names, work companies, job positions, highest educational level, and their work experience in the industries relating to food-service or teaching courses.

### 3.2. Participant Experts

The nature of the research topic was based mainly on the respondents' experience, authority, and willingness to cooperate. As for the number of experts, Delbecq et al. [34] suggested it should be 15–30 for the homogeneous group, and 5–10 for the heterogeneous group. A total of 20 experts of the food-service industry, government, and academic institutions were invited to participate. These experts had over 10 years of experience in food safety and hygiene supervision. Among them, three were hygiene officers in government, with 10-15 years of supervision, five were university teachers who had taught food-service hygiene courses for the past 10 years, and 10 were supervisors or directors of food safety and hygiene departments, including restaurants, independent canteens, chain restaurants, and central kitchens.

### 3.3. Data Analysis

After the collection of the MDM questionnaires, the first step was to examine whether or not there were any missing items, to ensure that the data were complete and correct. To verify the consistency of the experts' opinions, the average mean, standard deviation, and interquartile range were used as assessment criteria. According to the mean averages, the indicators were categorized according to their importance [35]. The average mean implied the importance of each indicator. When the mean was ≥3.5, it showed that the indicator was considered to be important by Delphi experts. When there was a standard deviation > 1, it meant that there was a high numerical discretization and divergence of the experts' opinions. The results with an interquartile range of ≤0.5 implied that the experts' opinions on the indicators reached a high consistency [36]. Linstone and Turoff [35] considered that a change of less than 15% for each item, between the two surveys and evaluated by all experts, indicated stability. The alternative option is total stability, which describes the mean stability of all the items that equaled or exceeded 70%.

## 4. Results and Discussion

### 4.1. Comparison of Three Rounds of Delphi Questionnaire Survey

In terms of the main indicators of the scale of culinary professional competence of kitchen staff for the use of surplus food, the results of the three rounds were that the competence of attitude was the most important, followed by knowledge and skills, and when dealing with the problems and difficulties of surplus food, the importance of all of them reached 4 or more points (Table 1).

**Table 1.** Statistical table of expert opinions on the scale of culinary professional competence of kitchen staff for the use of surplus food.

| Main Indicators | Round 1 | Round 2 | Round 3 |
|---|---|---|---|
| Item | Mean | Mean | Mean |
| C1. Knowledge competence of surplus food | 4.77 | 4.68 | 4.64 |
| C2. Attitude competence of surplus food | 4.82 | 4.82 | 4.73 |
| C3. Skills competence of surplus food | 4.45 | 4.59 | 4.55 |
| C4. Problems and difficulties of surplus food | 4.50 | 4.55 | 4.36 |

### 4.2. Analysis of the Three Round of the Delphi Questionnaire Survey

Tables 2–5 are the statistical tables of the expert opinions of the second round of the Delphi questionnaire survey. According to these tables, it can be seen that the average mean of all the indicators was greater than 4.00; in addition, the interquartile range of all indicators was less than,

or equal to, 1, which did not meet the criteria for deleting items (which require that the mean is less than 4.00 and the interquartile range is greater than, or equal to, 1), and the stability was more than 85%, so that all indicators were retained. The means were all above 4.00 for the main indicators of "C1: Knowledge competence of surplus food handling (mean 4.64)", "C2: Attitude competence of surplus food handling (mean 4.73)", "C3: Skill competence of surplus food handling (mean 4.55)", "C4: Dealing with the problems and difficulties of surplus food (mean 4.36)", where the attitude competence was the most important, followed by the knowledge and skills competencies. The most important items of the knowledge competence were "K8: Know how to store various ingredients", "K6: Have the concept of FIFO(First In First Out)", and "K10: Store ingredients in accordance with washing procedures or different times and areas to avoid decomposition, due to the contamination of ingredients" (Table 2). The important items of attitude competence (Table 3) were "A6: Pay attention to the freshness of food and storage temperature", "A1: Have the idea of not wasting food," "A8: Classify the management of the preservation of ingredients." "S13: Follow the rule of first-in, first-out (FIFO)," "S19: Effectively plan storage and storage spaces for refrigerators," and "S9: I take into account the characteristics of the ingredients when designing the menu" were the most important factors in skill competence (Table 4). "Q8: Strengthen (re-educate) the public's concept of reuse of surplus food," "Q3: The overpacking of ingredients can also cause waste of resources," and "Q7: The government should promote the methods for the disposal of surplus food and provide channels" were the most important problems to solve (Table 5).

**Table 2.** Results of knowledge competence indicators.

| Professional Competences for Knowledge | | | | | |
|---|---|---|---|---|---|
| Item | Mean | Mode | Interquartile Range | Stability | Priority Order |
| K1. Understand the seasons and prices of various ingredients | 4.50 | 5 | 1.00 | 90.9 | 11 |
| K2. Know how to control ingredients from the source | 4.82 | 5 | 0.00 | 100.0 | 4 |
| K3. Understand the characteristics of the ingredients before designing the menu | 4.73 | 5 | 0.25 | 95.5 | 5 |
| K4. Have good food characteristics knowledge for kitchen staff | 4.68 | 5 | 1.00 | 100.0 | 8 |
| K5. Have the concepts of menu design and setting up meal sizes | 4.55 | 5 | 1.00 | 100.0 | 10 |
| K6. Have the concept of first-in, first-out (FIFO) | 4.91 | 5 | 0.00 | 100.0 | 2 |
| K7. Know the proper packaging or sub-packaging size for ingredients | 4.23 | 4 | 0.25 | 100.0 | 13 |
| K8. Know how to store various ingredients | 5.00 | 5 | 0.00 | 100.0 | 1 |
| K9. Understand the characteristics of ingredients for pre-treatment | 4.68 | 5 | 1.00 | 100.0 | 8 |
| K10. Store ingredients in accordance with washing procedures or different times and areas to avoid decomposition, due to the contamination of ingredients | 4.91 | 5 | 0.00 | 100.0 | 2 |
| K11. Be familiar with how to cook ingredients | 4.73 | 5 | 1.00 | 100.0 | 5 |
| K12. Understand how food is sterilized, to reduce pollution | 4.73 | 5 | 1.00 | 100.0 | 5 |
| K13. Learn about government regulations related to food and surplus | 4.41 | 4 | 1.00 | 100.0 | 12 |

**Table 3.** Results of attitude competence indicators.

| Professional Competences for Attitudes | | | | | |
|---|---|---|---|---|---|
| Item | Mean | Mode | Interquartile Range | Stability | Priority Order |
| A1. Have the idea of not wasting food | 4.86 | 5 | 0.00 | 100.0 | 2 |
| A2. Ability to adhere to conservation principles (cooking or saving energy) | 4.55 | 5 | 1.00 | 100.0 | 8 |
| A3. Personal morals tell me not to waste | 4.45 | 5 | 1.00 | 95.5 | 10 |
| A4. I always think of the interest of the operator when designing the menu, to save ingredients | 4.18 | 4 | 1.00 | 86.4 | 13 |
| A5. Supervision, to avoid waste of ingredients | 4.68 | 5 | 1.00 | 95.5 | 5 |
| A6. I will pay attention to how fresh food is stored | 4.95 | 5 | 0.00 | 100.0 | 1 |
| A7. Pay attention to the package integrity of ingredients | 4.68 | 5 | 1.00 | 100.0 | 5 |
| A8. Classify management of the preservation of ingredients | 4.86 | 5 | 0.00 | 100.0 | 2 |
| A9. I discard kitchen waste in accordance with classification methods | 4.36 | 4 | 1.00 | 95.5 | 12 |
| A10. Use education to convey the idea of reducing surplus | 4.77 | 5 | 0.00 | 95.5 | 4 |
| A11. Culinary staff need to proactively learn (e.g., using new tools on how to process ingredients) | 4.45 | 5 | 1.00 | 95.5 | 10 |
| A12. Surplus education or concepts are integrated into the daily work of culinary staff | 4.50 | 5 | 1.00 | 90.9 | 9 |
| A13. It is wrong to take kitchen ingredients home privately | 4.64 | 5 | 1.00 | 90.9 | 7 |

**Table 4.** Results of skills competence indicators.

| Professional Competences for Skill | | | | | |
|---|---|---|---|---|---|
| Item | Mean | Mode | Interquartile Range | Stability | Priority Order |
| S1. I have the ability to control ingredients from the source | 4.59 | 5 | 1.00 | 100.0 | 13 |
| S2. I have the ability to quantitatively use ingredients and raw materials | 4.73 | 5 | 0.25 | 95.5 | 8 |
| S3. I use seasonal ingredients | 4.55 | 5 | 1.00 | 95.5 | 15 |
| S4. I choose the right raw materials and logistics providers | 4.59 | 5 | 1.00 | 100.0 | 13 |
| S5. I can accurately calculate the purchase quantity | 4.36 | 4 | 1.00 | 90.9 | 18 |
| S6. Effectively control the daily consumption of ingredients | 4.64 | 5 | 1.00 | 95.5 | 10 |
| S7. I know how to package food properly | 4.23 | 4 | 1.00 | 81.8 | 21 |
| S8. Ability to use tools without wasting ingredients (e.g., scraping off residue) | 4.00 | 4 | 0.00 | 81.8 | 23 |
| S9. I take into account the characteristics of the ingredients when designing the menu | 4.82 | 5 | 0.00 | 95.5 | 3 |
| S10. Pay attention to the skills and knowledge of chefs when designing the menu | 4.50 | 5 | 1.00 | 95.5 | 17 |
| S11. I use surplus to increase its added value, such as making full use of it to cook various employee meals or make soup | 4.55 | 5 | 1.00 | 95.5 | 15 |
| S12. I make the most of the ingredients | 4.64 | 5 | 1.00 | 95.5 | 10 |
| S13. Follow the rule of first-in, first-out (FIFO) | 5.00 | 5 | 0.00 | 100.0 | 1 |
| S14. I know the procedures when using ingredients, so that there is no repeated thawing | 4.82 | 5 | 0.00 | 95.5 | 3 |

**Table 4.** *Cont.*

| Professional Competences for Skill | | | | | |
|---|---|---|---|---|---|
| **Item** | **Mean** | **Mode** | **Interquartile Range** | **Stability** | **Priority Order** |
| S15. Follow storage standards | 4.82 | 5 | 0.00 | 100.0 | 3 |
| S16. Have capacity for inventory control | 4.82 | 5 | 0.00 | 100.0 | 3 |
| S17. Warehouse classification management | 4.64 | 5 | 1.00 | 100.0 | 10 |
| S18. Avoid food damage caused by environmental pollution | 4.82 | 5 | 0.00 | 100.0 | 3 |
| S19. Effectively plan storage and storage spaces for refrigerators | 4.86 | 5 | 0.00 | 100.0 | 2 |
| S20. Increase the shelf-life of food by processing methods | 4.14 | 4 | 1.00 | 81.8 | 22 |
| S21. Have the ability to observe kitchen waste to improve the menu | 4.68 | 5 | 1.00 | 95.5 | 9 |
| S22. Have the ability to flexibly adjust ingredients and menus | 4.27 | 4 | 1.00 | 86.4 | 20 |
| S23. Can improve the menu through sales data | 4.36 | 4 | 1.00 | 95.5 | 18 |

**Table 5.** Results of problem and difficulty items.

| Problem and Difficulty | | | | | |
|---|---|---|---|---|---|
| **Item** | **Mean** | **Mode** | **Interquartile Range** | **Stability** | **Priority Order** |
| Q1. Surplus causes social problems | 4.23 | 4 | 1.00 | 90.9 | 6 |
| Q2. The current classification of kitchen waste is unclear | 4.18 | 4 | 1.00 | 86.4 | 7 |
| Q3. The overpacking of ingredients can also cause the waste of resources | 4.64 | 5 | 1.00 | 95.5 | 2 |
| Q4. Surplus causes poor environmental health | 4.09 | 4 | 0.00 | 90.9 | 9 |
| Q5. Bringing kitchen ingredients home without permission is improper | 4.14 | 4 | 1.00 | 86.4 | 8 |
| Q6. At present, government units have not formulated regulations related to surplus food | 4.27 | 4 | 1.00 | 95.5 | 5 |
| Q7. The government should promote the methods for the disposal of surplus food and provide channels | 4.64 | 5 | 1.00 | 100.0 | 2 |
| Q8. Strengthen (re-educate) the public's concept of re-use of surplus food. | 4.73 | 5 | 1.00 | 100.0 | 1 |
| Q9. To reduce the waste of surplus food, the requirements and formulation of relevant laws and regulations must be carefully studied | 4.36 | 4 | 1.00 | 95.5 | 4 |

*4.3. Discussion on Delphi Questionnaire Survey*

Tables 2–5 showed that the average means of all four of the main indicators were above 4.00, indicating that they were all important in the indicator of the kitchen staff's professional culinary functions, from the perspective of surplus food. The Workforce Development Agency [37] divided competencies into the following: (a) knowledge, which means to understand the general principles and facts that can be applied to the field in the implementation of a certain thing; (b) skill, which belongs to the cognitive ability or technical operation ability to perform a certain thing, or to help the task and the ability for personal-related communication, social interaction, and self-management behaviors; and (c) attitude, which belongs to the perception of things and the behavioral motivations that are adopted. Suhairom et al. [25] indicated that competencies are divided into two categories, namely, the one is a visible competence, including the technical and non-technical capabilities, and the other is an invisible competence that includes concepts, physical and mental states, physiological factors, and behaviors, which, in turn, affect work performance and situational performance. When the two performances interact, the personal work performance is judged. In summary, all the above

professional competencies will be influenced by the level of knowledge, attitude, and the skills of the individual in the discussion of the indicators of the kitchen staffs' professional competence in the culinary arts, from the perspective of surplus food.

Experts from all walks of life believe that if the indicators of the kitchen staffs' professional culinary competencies are discussed from the perspective of surplus food only, then dealing with the problems and difficulties of surplus food are slightly insufficient. The USEPA [38] stated that the U.S. Environmental Protection Agency (EPA), the United States Department of Agriculture (USDA), and the US Food and Drug Administration (FDA) have jointly launched a Winning on Reducing Food Waste Federal Inter-agency Strategy to strengthen cooperation, coordination, and education, in order (1) to strengthen inter-agency coordination, (2) to implement consumer education, (3) to strengthen coordination and guidance on food waste standards, (4) to clarify and to convey information on food safety, food expiration labeling, and food donation, (5) to cooperate with private companies in reducing food loss and waste throughout the supply chain, and (6) to encourage federal agencies to reduce food waste in their respective catering areas. Wang et al. [39] pointed out that the concept of reducing food waste needs to be implemented by educating kitchen staffs and consumers on how to reduce food waste. The New Taipei City Surplus-Food Network [40] encourages the public to cultivate the concept of cherishing food through advocacy and childhood education, so that any surplus food, or food that is not needed at all, will be donated directly to enable people in need to enjoy the true value of food. In summary, for "C4: Dealing with the problems and difficulties of surplus food", not only do the kitchen staff need to have the knowledge, attitude, and skill themselves, but these also need to be promoted by the government, and education on how to cherish food must be enhanced to help reduce the problem of food waste through the joint efforts of society, as a whole.

## 5. Conclusion and Suggestions

### 5.1. Conclusions

This study used interviews and modified the Delphi method to modify the content of the interviews and three-round expert questionnaires to form four indicators, including the knowledge competence indicator of 13 items, the attitude competence indicator of 13 items, the skills competence indicator of 23 items, and the problem and difficulty indicator of 9 items.

The following are the knowledge indicators, in terms of their importance: "K8: Know how to store various ingredients," "K6: Have the concept of FIFO," "K10: Store ingredients in accordance with washing procedures or different times and areas to avoid decomposition, due to contamination of ingredients", were the most important factors," while "A6: Pay attention to the freshness of the food and the storage temperature," "A1: Have the idea of not wasting food," and "A8: Classify the management of the preservation of ingredients" were the most important attitude factors. For the importance of skill competence analysis, "S13: Follow the rule of first in, first out (FIFO)," "S19: Effectively plan storage and storage spaces for refrigerators," and "S9: I take into account the characteristics of the ingredients when designing the menu" were most important.

The experts believe that "Q8: Strengthen (re-educate) the public's concept of reuse of surplus food," "Q3: The overpacking of ingredients can also cause waste of resources" and "Q7: The government should promote the methods for the disposal of surplus food and provide channels" were the problem factors that need the most consideration at present.

### 5.2. Research Suggestions

#### 5.2.1. Suggestions to Kitchen Staffs

The attitude competence is the most important for developing the concept of surplus food and avoiding food waste. Chefs should pay more attention to the first-in-first-out principle, the freshness of the ingredients, and the effective use of storage space when handling ingredients, as these can

reduce food waste before food preparation. In addition, the menu design is also very important for the effective use of ingredients, as it can increase the utilization rate of food. Therefore, this study suggests that self-learning, company education and training, online media, government regulations, skills certification, on-the-job training, etc. can be used as a basic channel for the kitchen staff to strengthen their concept of surplus food.

### 5.2.2. Suggestions to the Government

This study shows that the government should advocate the treatment of surplus food, that it should provide channels for reducing the waste of surplus food, and that it should carefully study the requirements and formulation of the relevant laws and regulations. If the current government had the complete legal requirements it would strongly advocate for the treatment methods and channels of surplus food, and it would be able to tackle the issues related to surplus food.

The government is urged to develop laws and regulations related to surplus food, to charge kitchen waste treatment fees for catering businesses and the general public, and to vigorously examine the fine parts of food waste (food waste, compost), in order to effectively reduce or deter food waste. In addition, the government should develop the standards and specifications for the use or donation of soon-to-expire food and the food that can be safely used, and it should also cooperate with the catering industry for more catering promotion. Local governments are urged to advocate for the disposal of surplus food to the public, and to use seminars, posters, advertisements, online multiple media, and other methods for its effective communication with the public. Channels should also be established for local governments to plan and deal with surplus food, such as food banks, or shops for below-standard fruit and vegetables.

### 5.2.3. Suggestions to Restaurants

Based on the results of knowledge, attitude, skills, and problem handling indicators of this study, it is recommended that restaurants should establish clear and standard operating procedures for ingredient procurement, packaging, preservation, picking, washing, cutting, cooking, and food waste, that they should strictly control the level of the material loss of food in each link, and use the methods of supervision, education, and training for food handling.

**Author Contributions:** Conceptualization, W.-H.K.; methodology, W.-H.K. and M.-Y.L.; data analysis, M.-Y.L. funding acquisition, W.-H.K.; investigation, M.-Y.L.; project administration, W.-H.K.; resources, W.-H.K.; supervision, W.-H.K.; validation, W.-H.K.; writing—original draft, W.-H.K. and M.-Y.L.; writing—review and editing, W.-H.K. All authors have read and agreed to the published version of the manuscript.

**Funding:** The authors acknowledge the financial support of the Ministry of Science and Technology, Taiwan. Project number MOST 108-2511-H-030 -002 -MY2.

**Conflicts of Interest:** The authors declare no conflict of interest.

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
