# Peer review of "Evaluation of the Professional Competence of Kitchen Staff to Avoid Food Waste Using the Modified Delphi Method"

_sustainability, doi:10.3390/su12198078_

Round 1

Reviewer 1 Report

The following should receive attention:

  • p. 2: last sentence of Introduction does not make sense.
  • p. 2: 2nd paragraph under Food waste: replace "eaten up" with consumed
  • p. 3: Methodology: 3.1 Declare the meaning of MDM
  • p. 4: Results and Discussion: last sentence on p4: should be Tables 2-6, as well as on p. 5 section 4.3
  •  I would recommed that the authors refer to each applicable table in the text and not only summarize Tables 2-6 explain...., e.g. when explaining the C factors, explain what it means, e.g. C = competence and refer to Table 2; when discussing knowledge, explain that K represents Knowledge and refer to Table 4. The also applies to A = Attitude = Table 4; S = Skills = Table 5; Q = Problem and difficulty = Table 6. Also indicate these in the Table itself or as a subscript or footnote. Consider to integrate the Tables into the text for clarity.

Reviewer 2 Report

The paper presents important problem that is definitely worth raising.

However, the literature review performed by the Authors could be more developed. 

The Authors obtained very interesting results presented in tables, but in my opinion they deserve wider, more detailed discussion, also with the results of other researchers. 

The whole article seems to be very brief and should be more developed.

Eventually, the references in the text are not presented according to the "Instructions for Authors".

Reviewer 3 Report

The paper presents an application of the well-known Delphi method to catering industry. It is well written, even if with some methodological weakness (in particular, a 5-point Likert scale seems to be very narrow, when passing to statistical summaries), but only some minor observations are to be made.

- Please, any time that a new acronym is introduced, write in full the expression to which it refers (for example, in the Introduction, write ‘Waste and Resources Action Programme’ in full, before introducing the acronym WRAP)

- Is “structural questionnaire” correct? Or should it be “structured questionnaire”, to indicate a well-defined set of questions?

- The concept of “average mean” (Section 3.3 and others) is a bit obscure, and most likely redundant: if it is the classic sample estimator of an (unknown) expected value, the term “mean” is sufficient.

- Table 1 is present twice, once in page 4 and once in page 6. Please renumber correctly the Tables, in increasing order.

- The citations from Linstone and Turoff (2002) are not correct: they are the editors of the volume, but the methods for the evaluations of stability seem to be taken from a paper within the volume, written by other authors: M. Scheibe, M. Skutsch, and J. Schofer, Experiments in Delphi Methodology, pages 257-281 of the volume edited by Linstone and Turoff.
